# Poly(Butylene Adipate/Terephthalate-Co-Glycolate) Copolyester Synthesis Based on Methyl Glycolate with Improved Barrier Properties: From Synthesis to Structure-Property

**DOI:** 10.3390/ijms231911074

**Published:** 2022-09-21

**Authors:** Yanning Wang, Haicun Yang, Bingjian Li, Shi Liu, Mingyang He, Qun Chen, Jinchun Li

**Affiliations:** 1School of Materials Science and Engineering, Changzhou University, Changzhou 213164, China; 2Jiangsu Key Laboratory of Advanced Catalytic Materials and Technology, Changzhou University, Changzhou 213164, China; 3Jiangsu Key Laboratory of Environmentally Friendly Polymeric Materials, National-Local Joint Engineering Research Center of Biomass Refining and High-Quality Utilization, Changzhou University, Changzhou 213164, China

**Keywords:** polycondensation, methyl glycolate, glycolic acid, barrier properties, hydrophilicity

## Abstract

The main problem of manufacturing with traditional biodegradable plastics is that it is more expensive than manufacturing with polymers derived from petroleum, and the application scope is currently limited due to poor comprehensive performance. In this study, a novel biodegradable poly(butylene adipic acid/terephthalate-co-glycolic acid) (PBATGA) copolyester with 25–60% glycolic acid units was successfully synthesized by esterification and polycondensation using cheap coal chemical byproduct methyl glycolate instead of expensive glycolic acid. The structure of the copolyester was characterized by ATR-FTIR, ^1^H NMR, DSC, and XRD; and its barrier property, water contact angle, heat resistance, and mechanical properties were tested. According to the experiment result, the PBATGA copolyesters showed improved oxygen (O_2_) and water vapor barrier character, and better hydrophilicity when compared with PBAT. The crystallization peaks of PBATGAs were elevated from 64 °C to 77 °C when the content of the GA unit was 25 mol %, meanwhile, the elongation at the break of PBATGA25 was more than 1300%. These results indicate that PBATGA copolyesters have good potentiality in high O_2_ and water vapor barrier and degradable packaging material.

## 1. Introduction

Nowadays, biodegradable materials have become a research hotspot in the field of materials science. They have attracted extensive attention due to their advantages in environmental protection and resource conservation, and the research on this material is also increasing [1,2,3]. Poly(butylene adipate terephthalate) (PBAT) is a kind of the most active biodegradable plastics due to its comprehensive properties close to that of general-purpose plastic polyethylene in the current market. Its production capacity has rapidly increased. According to chemical market information, it is predicted that the domestic PBAT capacity will likely exceed 2 million tons by 2025, with a 43% compound annual growth (CAGR) in capacity. Although PBAT has the largest application share in replacing traditional film bags and packaging [4,5], due to its low modulus [6], poor weather resistance [7], poor mechanical performance and barrier performance [8,9,10,11], and poor opening during the processing of blown film, PBAT cannot meet the requirements of the agricultural film. To further expand the application field of PBAT, it is particularly necessary to modify it.

At present, the modification method of PBAT is mainly blending modification, such as polylactic acid (PLA) [2,12,13,14,15,16,17,18,19], polybutylene succinate (PBS) [20], polyglycolic acid (PGA) [21,22,23,24,25,26,27], polycaprolactone (PCL) [28,29], poly(propylene carbonate) (PPC) [30,31,32,33], poly(ethylene-co-vinyl alcohol) (EVOH) [34], starch [35,36], calcium carbonate [37,38], and other materials [39]. PBAT was also modified by chain extension modification [40].

PGA, also known as polyglycolide or polyglycolic acid [41], is the simplest linear aliphatic polyester with good biodegradability and biocompatibility [42]. Among all the current biodegradable polyesters, its gas barrier properties and heat resistance properties are the best, and its flexural strength is also the highest [43,44]. However, its thermal decomposition temperature is low, and the processing temperature range in practical applications is very narrow, between 230 °C and 240 °C, which is not conducive to processing applications. Given the unique properties and potential price advantage of PGA, blending or copolymerizing modification of PGA and PBAT has great application potential in the field of disposable products such as heat-resistant tableware, degradable film bags, and barrier packaging. In view of the unique performance and potential price advantages of PGA, it is expected that novel PBAT/PGA blends or copolymers can be obtained via blending or copolymerization, which can be used for disposable consumer goods such as heat-resistant tableware, degradable film bags, and barrier packaging.

There have been some reports on the introduction of glycolic acid units into traditional biodegradable plastics. Glycolic acid units were introduced into PBF, and high molecular weight poly(butylene furandicarboxylate-co-glycolate) copolyester (PBFGA) was prepared by melting polycondensation and melt transesterification [45]. J. Zhu prepared the random copolyester poly(butylene-succinate-co-glycolate)(PBSGA) again by the same method and studied the effect of glycolic acid content on the thermal, mechanical, barrier properties, and biodegradability of the copolyester. The results showed that when the content of GA was 40%, the weight loss of PBSGA in seawater exceeded 22% after 49 days [46]. Subsequently, J. Ji synthesized PBFGA with GA units ranging from 20% to 50% by the above method. It was found that the PBFGA copolyesters showed an amorphous structure and had good hydrolysis properties, and when it was put in deionized water for 84 days, the weight loss rate of PBFGA50 exceeded 30%, and PBFGA copolyester had the potential to be a promising seawater degradable material [47].

However, because the monomer glycolide for synthesizing PGA is very expensive and the ring-opening polymerization is difficult, it is challenging to achieve mass production of PGA on a large device. Another polymer, polymethyl glycolate (PMG)—which has the same repeating unit as PGA—has both degradable properties and higher thermal stability. Its monomer methyl glycolate is a byproduct of the coal chemical industry, inexpensive, and readily available. If PMG is used instead of PGA and PBAT for modification, on the one hand, the byproduct resources are fully utilized and the cost is reduced; on the other hand, the hydrolysis rate and barrier properties, strength, and modulus of PBAT are improved. However, most of the existing technologies use blending, the system is complicated, the compatibility leads to limited mechanical properties, and corresponding additives need to be added, which affects biological safety.

In this essay, the byproduct methyl glycolate monomer of the coal chemical industry was introduced as one of the raw materials, and then the glycolic acid unit was incorporated into the main chain of PBAT by melting polycondensation from the perspective of molecular structure design and cost reduction, and a novel type of copolyester called poly(butylene/adipic terephthalate-co-glycolic) copolyester has been developed. Herein, we solved the problem of the waste of coal chemical resources. Furthermore, we broke through the technical bottleneck of polyglycolic acid, such as high price, high brittleness, narrow processing temperature range, limited application fields, and excessive degradation speed. More importantly, the problem of low barrier properties of dibasic acid diol copolyesters has also been solved.

## 2. Results and Discussion

### 2.1. FTIR and ^1^H NMR Characterization of PBATGAs

The chemical structures of PBAT and PBATGA copolyesters were detected by ATR-FTIR. As shown in Figure 1, PBAT and PBATGA copolyesters have almost identical IR spectra. Stretching vibration peaks of -CH_2_- was around 2961 cm^−1^. A strong peak at 1710 cm^−1^ was the C=O stretching vibrations, bending vibrations of -CH_2_- appeared at 1410 cm^−1^ and 1389 cm^−1^; and the absorption peaks at 1267 cm^−1^, 1167 cm^−1^, 1119 cm^−1^, and 1103 cm^−1^ were the C-O-C bonding. The peak at 725 cm^−1^ was relative to the C-H bending vibration of the benzene groups in BT segments. The shoulder of the absorption peak of PBATGAs at about 1710 cm^−1^ of C=O was broader than PBAT, suggesting that GA units are incorporated successfully in the copolymerization reaction. As shown in Figure 1b,c, the absorption peak area of PBATGAs at about 2961 cm^−1^ of -CH_2_- became smaller, and the absorption peak area of PBATGAs at about 725 cm^−1^ of C-H from para-substituent of the benzene ring was larger, suggesting the influence of GA group on -CH_2_- and benzene ring.

The molecular weight of PBATGAs was tested by GPC. As displayed in Table 1, it could be found that number average molecular weight (Mn) ranging from 29,758 to 61,847 g/mol and weight average molecular weight (Mw) ranging from 55,225 to 109,126 g/mol were obtained with polydispersity (PDI = Mw/Mn) from 1.70 to 1.91, as expected for the synthetic methods used. The results showed that the copolyesters synthesized by this method possessed high molecular weight.

Nuclear magnetic resonance is one of the commonly used methods to characterize the structure of polymer segments. Different hydrogen atoms can show different chemical shifts. Figure 2 showed the ^1^H NMR spectra of PBAT and PBATGA copolyesters. The chemical shift peak at 7.26 ppm was solvent deuterated chloroform. For PBAT, the chemical shift 8.08 ppm (a) was attributed to -C_6_H_6_- of BT units, 4.41 ppm (c) and 1.95 ppm belongs to -OCH_2_- and -CH_2_- of 1,4-BDO from TBT units respectively, 4.36 ppm (d) represented the -OCH_2_- of 1,4-BDO from TBA units, 4.12 ppm (e) and 4.07 ppm (f) belongs to -OCH_2_- of 1,4-BDO from BA unit in TBA units and ABA units respectively, 2.31 ppm (g) belongs to O=C-CH_2_- of adipic acid from BA unit. The signals of CH_2_ in GA units were located at 4.6–5.1 ppm. New chemical shifts appeared at 5.01 ppm (b_1_), 4.89 ppm (b_2_), 4.74 ppm (b_3_), 4.70 ppm (b_4_), and 4.61 ppm (b_5_) standing for the methylene in glycolate units connected with different units, indicating the glycolic acid groups were successfully copolymerized into the PBAT copolyester chain. In addition, the methylene proton peak of butanediol on the BA unit showed three new peaks (e_1_, e_2_, e_3_) at 4.2–4.5 ppm due to the influence of the glycolic acid group. The details were shown in Figure 2b. n_GA_ can be determined by Equation (1), the number-average sequence length of BA, BT, and GA (L_n, BA_, L_n, BT_, and L_n, GA_) could be calculated using Equations (2)–(4), respectively. The randomness (R) along with the number-average sequence length of BA, BT, and GA units were calculated using Equation (5). The ^1^H NMR data and the calculated results of other copolyesters were displayed in Table 1. It can be seen that the L_n__, GA_ of the copolyesters are between 1.5 and 2.15, and the value of L_BS_ and L_BT_ changed dominantly with the compositions of copolyesters, which is logical. The R value is 0.9 when n_GA_ is less than 35%, indicating that the obtained polymer is a block copolyester. The R value is 1.09–1.19 when n_GA_ is 35–40%, implying that the GA units exist in an alternating form in the PBAT molecular chain. The R value is greater than 1 when n_GA_ is greater than 50%, which is inferred that random copolyesters were successfully synthesized.
(1)$$n_{n,\mathrm{GA}}=\frac{{2I}_{b}}{I_{a}+2I_{b}}$$
(2)$$L_{n,\mathrm{BA}}=\frac{1}{4}\left(\frac{I_{f}+I_{e4}+{2I}_{g1}}{I_{g1}}+\frac{I_{e3}}{I_{b5}}\right)$$
(3)$$L_{n,\mathrm{BT}}=\frac{1}{4}\left(\frac{I_{c}+I_{d}+{2I}_{e3}}{I_{e2}}+\frac{I_{a}}{I_{b1,b2}}\right)$$
(4)$$L_{n,\mathrm{GA}}=\frac{1}{3}\left(\frac{I_{b1,b2}+I_{b3,b4}}{I_{b1,b2}}+\frac{I_{g1}+I_{b3,b4}}{I_{g1}}+\frac{I_{e3}+I_{b3,b4}}{I_{e3}}\right)$$
(5)$$R=\frac{1}{L_{n,\mathrm{BA}}}+\frac{1}{L_{n,\mathrm{BT}}}+\frac{1}{L_{n,\mathrm{GA}}}$$

### 2.2. Crystallization Behavior and Crystal Structure

The crystallization behavior of PBAT and PBATGAs was measured by DSC. The collected DSC data were processed to obtain the corresponding crystallization parameters, mainly including peak temperature of crystallization (T_c_), the difference between the topset crystallization temperature (T_opset_) and T_c_, the glass transition temperature (T_g_), the exothermic enthalpy of crystallization (ΔH_c_), the melting temperature of crystals (T_m_), and the melting enthalpy(ΔH_m_); the relevant information are presented in Table 2, the crystallinities during the cooling process (X_c_) were calculated according to Equation (6):(6)$$X_{c}(\%)=\frac{{\Delta H}_{c}}{{\Delta H}_{m}^{0}}\times 100\%$$
where ${\Delta H}_{m}^{0}$ is the melting enthalpy of 100% crystallized PBAT is the enthalpy of 114 J/g.

In the cooling scan (Figure 3a), Tc and ΔH_c_ of PBAT were 64 °C and 35.53 J/g, respectively. PBATGA25 showed that the crystallization peaks around 77 °C and the exothermic enthalpy of crystallization is 21.47 J/g, relative to an increase of 13 °C compared to PBAT. Meanwhile, a faster crystallization rate can be observed because it has a smaller value of T_opse__t_-T_c_, which can be used as a judgment for the overall rate of crystallization. This value was about 11.98 °C when the GA content was 25%, which was lower than that of neat PBAT (15.55 °C), indicating that the proper introduction of glycolic acid into PBAT could increase its crystallization temperature and improve its crystallization rate; this can be ascribed to the polarity between GA molecular chains being relatively strong, which enhances the intermolecular force, making the intermolecular arrangement more orderly and easier to crystallize [48]. However, when further raising the molar content of GA to 35%, the crystallization peak temperature of the copolyester dropped to 50.52 °C, which was 13.89 °C lower than that of PBAT. When the molar content of glycolic acid was 40–60%, there was no crystallization peak in the DSC spectrum of the copolyester, indicating that the addition of glycolic acid destroyed the crystallization of PBAT and made the copolyester show an amorphous state. It was noticed that variations of GA units influenced the crystallization of PBAT significantly. A small quantity of GA units can improve the crystallization of PBAT. A mass of GA units can inhibit the crystallization of PBAT. In summary, PBATGAs can be classified into three types depending on their crystallization behavior: PBATGA25 (good), PBSGA30–PBATGA40 (middle), and PBATGA50–PBATGA60 (weak). In the second heating scan (Figure 3b), T_m_ and ΔH_m_ of PBAT were 129.9 °C and 21.02 J/g, respectively. It can be seen that the addition of 40–50% glycolic acid makes the copolyester show cold crystallization behavior. T_m_ decreased with increasing n_GA_, the melting point, and melting enthalpy of PBATGAs from 129.4 °C to 90.29 °C and from 31.57 to 13.07 J/g, respectively; indicating that the lamellar thickness was suppressed by the GA units. However, Tg showed the opposite trend (−27.02 °C to −18.05 °C).

We further explored the crystallization properties of PBAGAs copolyester by using the WAXD device. The WAXD curves of PBATGAs were shown in Figure 4. The diffraction peaks of PBAT and PBATGAs were at 16.36° (011), 17.55° (010), 20.50° (101), 23.25° (100), and 25.00° (111), respectively. The intensity of crystalline diffraction peaks gradually weakens with the increase in GA content, which suggests that the content of crystalline structure in PBATGAs decreases. It shows that the addition of glycolic acid will reduce the regularity of the main chain of PBAT, thereby reducing the crystallinity of the copolyester. In addition, although the crystal diffraction peak intensities of the PBATGAs are slightly different, the position of the crystal diffraction peak remains unchanged, which indicates that the addition of glycolic acid will not affect the formation of the PBAT crystal region and crystal structure.

### 2.3. Thermogravimetric Analysis

As we all know, a copolyester with good thermal stability is one of the keys to adapt to different molding processes. The thermal stabilities of PBATGAs in N_2_ atmosphere were evaluated by TG analysis. Figure 5a showed the thermal decomposition curves for PBATGAs and Figure 5b showed the DTG–T curves for PBATGAs, the corresponding data were summarized in Table 2. The relevant data showed that PBAT had higher thermal decomposition temperature than PBATGAs, the temperatures corresponding to 5% weight loss of the series samples (T_5%_) decreased from 376 °C to 350 °C, and the maximum decomposition temperature (T_dmax_) decreased from 418 °C to 411 °C with the increase in the GA segment. Thus, all PBATGAs will not decompose when the processing temperature is less than 300 °C. This ensures that the copolyester has good thermal stability during processing.

### 2.4. Barrier Properties

Oxygen permeability coefficients (P_O2_) and water vapor permeability coefficients (P_WV_) are listed in Table 3. The barrier improvement factor (BIF) is defined as the permeability coefficient of neat PBAT divided by those of the PBATGAs. The corresponding BIF results were displayed in Table 3. The values of BIF_O2_ and BIF_WV_ were greater than 1, indicating that PBATGAs has better oxygen and water barrier properties than PBAT.

In comparison with PBAT, the PBATGA25 and PBATGA40 copolyesters exhibited excellent oxygen and water barrier properties. In general, lower crystallinity results in lower oxygen barrier properties. However, the results of this study show that PBATGAs have low crystallinity, but their O_2_ barrier performance is significantly improved, so it has good performance advantages in this regard. Therefore, the presence of GA units is crucial for improving the oxygen barrier performance. In addition, the ester bond density of GA unit in PBATGA copolyesters is high, the polarity is stronger, the molecular chain is closely arranged, and the free volume is small compared with PBAT. The O_2_ barrier of the remaining PBATGA copolyesters reduced in oxygen permeability coefficients, mainly for the reason of its relatively lower molecular weight.

The water vapor barrier properties of other PBATGA copolyesters except for PBATGA35 have been improved, indicating the incorporation of GA units significantly improves the water vapor barrier properties of PBAT. This is due to the relatively high hydrophilicity of the GA units, and the water vapor molecules preferentially adsorb to the GA units, while the free volume of the GA units is small, so it will hinder the diffusion of water vapor molecules. In summary, the results verify the potential value of PBATGA copolyesters as high gas and water vapor barrier packaging material.

### 2.5. Surface Hydrophilicity

Hydrophilicity is an important factor affecting the degradation rate of degradable polyester materials. Its hydrophilicity can also directly reflect the strength of hydrolysis during its degradation since PBATGA copolyester is a biodegradable polyester. The water contact angle was tested, while solid materials can be classified into three categories based on the magnitude of this parameter: hydrophilic < 90°, hydrophobic at 90–150°, and superhydrophobic > 150°. Yue Ding et al. introduced GA units into the PBS homopolyester. The research results proved the increase in GA content improved the degradation rate of copolymer in acidic and neutral solution [49]. As shown in Figure 6, the GA units were beneficial for hydrophilicity. According to the above result, the water content angle declines gradually from 96 ± 1.0° to 78.5 ± 0.5° when the molar content of GA increases from 0% to 60%. This means that the increasing GA content has a positive correlation with hydrophilicity. PGA was a fast-hydrolyzing material in which the ester bond could be hydrolyzed quickly. Thus, copolymerization of PGA and PBAT could increase the hydrolysis rate of copolyester, and then accelerate its biodegradation rate. We speculated that by introducing the easily hydrolyzable GA segment into the PBAT chain, PBATGAs will have a faster degradation rate than PBAT.

### 2.6. Heat Resistance Analysis

The Vicat softening point (VSP), also known as Vicat heat resistance, is used to characterize the heat resistance of relative materials. The higher the Vicat softening point of a polymer material, the better its heat resistance is. Figure 7 showed the Vicat softening points of molded PBAT and PBATGA copolyesters. As can be seen from Figure 8, the VSP of PBATGA copolyesters tends to decrease as the GA content increases, and the heat resistance of PBATGA25 is comparable to that of neat PBAT. However, when the molar content of GA was 35%, the VSP of PBATGA copolyesters decreased to about 61.6 °C, a decrease of 26 °C compared to that of PBAT; when the molar content of GA was 50–60%, the VSP of PBATGA copolyesters decreased to below 60 °C; and it decreased the most when the molar content of GA was 60%, the VSP decreased to 51.9 °C, a decrease of nearly 36 °C. The decrease in VSP indicates that the addition of the glycolic acid group disrupts the regularity of the PBAT main chain, making it less susceptible to crystallization, as indicated by the drastically increased X_c_ values (22.22% to 1.32%), ultimately leading to a decrease in the heat resistance of its copolyesters.

### 2.7. Mechanical Properties

The tensile stress–strain curves of PBAT and PBATGA copolyesters were presented in Figure 8. The corresponding tensile strength and elongation at break were tested and shown in Table 4. From PBAT to PBATGA25, the stress and strain increased to 16–21 MPa and 1070–1304 MPa, respectively. The reason might be the higher crystallization temperature and faster crystallization rate of PBATGA25. For PBATGA35–PBATGA60, the tensile strength decreases from 7 to 2 MPa. In the meantime, the elongation at break decreases from 752% to 109%. This could be attributed to GA units increasing the randomness of the chains and increasing the movement ability of the PBATGA chains, as well as the degree of polymerization being difficult to increase when GA units continues to increase. When compared with other biodegradable polyesters, the toughness property of PBATGA25 (1304%) is higher than those of PPC, PBA_m_T55, PBSGA25, PBFGA20. It can be called the most promising biodegradable film bag raw material in the field of film and packaging.

## 3. Materials and Methods

### 3.1. Materials

1,4 Butanediol (BDO) was bought from Sinopharm Chemical Reagent Co., Ltd. (colorless transparent liquid, purity ≥ 99%, Shanghai, China). Terephthalic acid (TPA) was bought from Aladdin Chemicals Co. (white powder, purity ≥ 99%, Shanghai, China). Adipic acid (AA) was obtained from Shanghai Lingfeng Chemical Reagent Co., Ltd. (white powder, purity ≥ 99.5%, Shanghai, China). Methyl glycolate (MG) was purchased from Aladdin Chemicals Co. (colorless transparent liquid, purity ≥ 98%, Shanghai, China). Antimony trioxide (Sb_2_O_3_) (AR grade) was bought from Aladdin Chemicals Co. (purity ≥ 99.5%, Shanghai, China). Catalyst CAT-2019 (light yellow transparent viscous liquid) was supplied from the College of Advanced Materials Engineering, Jiaxing Nanhu University. Polymethyl glycolate (light yellow powder) was made in our laboratory. Tetrahydrofuran (AR grade) was purchased from Shanghai Lingfeng Chemical Reagent Co., Ltd. All of these reagents were used directly as received without further purification.

### 3.2. Synthesis

#### 3.2.1. Synthesis of Oligo(Methyl Glycolate) (OMG)

The weighed methyl glycolate and Sb_2_O_3_ (0.1 wt %) were added into a flask equipped with a stirrer. The system was heated to 160 °C for about 4–6 h until the methanol is no longer evaporated. Then the system was slowly evacuated by the water pump until the vacuum degree was 3000 Pa, the reaction was finished until a white solid was formed on the inner wall of the flask, and the stirring was stopped. The product was called oligo(methyl glycolate) (OMG).

#### 3.2.2. Synthesis of PBATGAs

As displayed in Figure 1, the poly(butylene/adipic terephthalate-co-glycolic) copolyester was synthesized in two stages: esterification and polycondensation. In the first stage, 1,4-butanediol, terephthalic acid, and adipic acid with CAT-2019 as catalyst were added to a 100 mL round-bottomed flask (AA/TPA molar ratio 50/50; diol/diacid 1.8/1). Transesterification was conducted at 200–230 °C for 4–5 h until more than 95% of the theoretical methanol was collected by a condenser. After this, the condenser tube and water separator were removed, OMG and Sb_2_O_3_ were added into the round-bottomed flask. Then the system was evacuated to a vacuum degree of less than 20 Pa. When the viscosity of the product was high or the phenomenon of climbing rods occurred (about 4–5 h), the reaction was stopped to obtain a PBATGA copolyesters as the final product. According to the molar percentage of glycolic acid units (n_GA_), the PBATGAs were labeled as PBATGA25, PBATGA35, PBATGA40, PBATGA50, and PBATGA60. The appearance of the final products was shown in Figure 9. Pure PBAT appeared bright milky white. The product appearance changes from milky white to light yellow with the increase of glycolic acid content, and the PBATGA60 copolyester emerged dark yellow, which is probably due to the thermal decomposition reaction of the partial oligomer methyl glycolate at 230 °C.

### 3.3. Characterization

FT-IR spectra were obtained by using an attenuated total reflectance–Fourier transform infrared (ATR-FTIR) spectrometer (IRAffinity-1S, Shimadzu Corporation, Shimadzu, Japan). The scan range was from 500 to 4000 cm^−1^ and each sample was in total reflection scanning mode.

The Mn, Mw, and dispersity (PDI) of PBATGAs were determined via GPC (AcquityAPC, Waters, MA, USA), tetrahydrofuran (THF) was used as the mobile phase with 10 mg of sample. It was filtered by a 0.45 µm filter after it was dissolved completely, the injection volume was 50 µL, the refractive index detector was the basic detector, and the column and detection temperature were both 35 °C.

The chemical structure was recorded by ^1^H NMR on a Bruker Avance NEO 600 (600 M) nuclear magnetic resonance instrument (Bruker corporation, Saarbrucken, Germany). ^1^H NMR was performed with TMS as the internal standard.

Differential scanning calorimetry (DSC) analysis was performed on a DSC 2500 instrument (TA Company, USA.) A sample with a mass of 10 mg was taken and put into an aluminum pan. The samples were first heated to 200 °C at 10 °C·min^−1^, kept at 200 °C for 3 min, and then they were cooled to −50 °C at 10 °C min^−1^ and keep 3 min, and reheated again to 200 °C at the same heating rate.

The X-ray diffraction (XRD) patterns were tested by a X-ray diffractometer (Aeris, PANalytical B.V., Netherlands), using a Cu Kα (=0.154 nm) radiation in the range of 5–60° at 5°min^−1^.

Thermogravimetric (TGA) analysis was carried out on a TGA5500 instrument (TA Company, Boston, MA, USA). In a nitrogen environment, the samples were heated to 800 °C at a speed of 20 °C·min^−1^. The temperature at 5% weight loss (T_d,5%_) and T_d,max_ were calculated.

The oxygen barrier properties of samples (thickness is 0.02 cm, 295 cm^2^) was detected with the pressure method at 23 °C under 0%RH conditions by using an oxygen transmission rate tester (GTR-V3) from Shandong Puchuang Industrial Technology Co., Ltd. (Weifang, China). The oxygen permeability coefficient (P_O2_) was obtained according to the GB/T 1038-2000.

The water vapor barrier properties of samples (thickness is 0.02 cm, 33 cm^2^) was measured with the weight gain and loss method at 23 °C under 50% RH conditions by using a water vapor transmission rate tester (WVTR-RC6) from Shandong Puchuang Industrial Technology Co., Ltd. (Weifang, China). The water vapor permeability coefficient (Pwv) was determined by following the GB/T 1037-2021.

The water contact angle was tested by using a contact angle meter (Data physics -OCA20, Germany). Three or five drops were observed on different areas for each film and contact angles were reported as the average value ± standard deviation.

Heat resistance properties were checked by ZWK1302-1 type heat deformation tester from Shenzhen Xinsansi Material Testing Co., Ltd. (Shenzhen, China). A detection method was used to conduct the test in accordance with the national standard GB/T 1633-2000. 10 (mm) × 10 (mm) × 4 (mm) was used for the sample strip, the heating rate was 120 °C/h and the weight added was 10 N.

The tensile properties of the plastic were tested based on GB/T 1040.3-2006 by WDT-10 (Shenzhen, KEJALI) universal testing machine. The samples were of dumbbell type with a total length of 75 mm, a standard distance of 20 mm, a thickness of 1 mm, and a tensile rate of 20 mm/min.

## 4. Conclusions

The main goal of the current study was to utilize methyl glycolate—a byproduct of the coal chemical industry—in the molecular backbone of PBAT through copolymerization. In this study, we successfully obtained high molecular weight poly(butylene adipate/terephthalate-co-glycolate) copolyester with tunable contents of glycolic acid units from methyl glycolate by esterification and polycondensation process. Based on these experimental results, the main conclusion can be summarized as follows: in comparison with the neat PBAT, PBATGA copolyesters containing 25 mol % GA units have higher crystallization temperature, tensile strength, and elongation at break. PBATGA copolyesters containing 35–60 mol % GA unit have weaker crystallizability, lower heat resistant temperature, and lower tensile strength. In addition, when the content of GA units is less than 60%, PBATGA is a semicrystalline copolyester, and its melting point is higher than 90 °C. PBATGA copolyester and PBAT keep the same crystalline structure and the decomposition temperature of PBATGA copolyester at 5% weight loss is more than 350 °C. Therefore, it has good thermal stability to guarantee reasonably good melt processability—even higher than that of neat PGA.

Most importantly, the PBATGA copolyesters manifest good hydrophilicity, and higher O_2_ and water vapor barrier properties. The PBATGA copolyesters may find uses in high-barrier biodegradable materials, such as replacing poly-vinylidene chloride (PVDC) or EVOH materials. In addition, tuning the crystallization behavior of PBAT by using the GA units will be explored. Efforts to study the shelf life of PBATGA copolyesters are very necessary in the future.

## Data Availability

The authors declare that the data supporting the findings of this study are provided in the main article and can be accessed upon request via email to the corresponding authors.

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
