# Peer review of "Poly(Butylene Adipate/Terephthalate-Co-Glycolate) Copolyester Synthesis Based on Methyl Glycolate with Improved Barrier Properties: From Synthesis to Structure-Property"

_ijms, 2022, doi:10.3390/ijms231911074_

Round 1

Reviewer 1 Report

The paper deals with very important problem of biodegradable plastics. The authors obtained high molecular weight Poly (bu- 365 tylene adipate/terephthalate-co-Glycolate) copolyester with tunable contents of glycolic 366 acid units from methyl glycolate by esterification and polycondensation process.

The paper is factually correct, but needs a revision:

1. Materials - please describe the materials that were bought - giving only manufacturers is not enough. Please write in what form and amount the ingredients were acquired. Did you check the composition and purity of them?

2. Please review the units (pa or Pa for vacuum).

3. FTIR analysis - fig. 2 - please analyses all the characteristic changes in the obtained curves. The same comment for Fig. 4 and 9.

4. For mechanical properties study - please show the samples, how many samples were tested, are the results repeatable - please discuss.

5. Conclusions – please explain why „Efforts to study the shelf life of PBATGA copolyesters 382 are very necessary for future”. Any future plans, especially for the results implementation? 

Reviewer 2 Report

Publication “Poly(butylene adipate / terephthalate-co-Glycolate) Copolyester Synthesized based on Methyl Glycolate with Improved Barrier Properties: from Synthesis to Structure-property” by Y. Wang et al. discusses synthesis and characterization of poly (butylene adipic acid / terephthalate-co-glycolic acid) (PBATGA) copolyesters with 25-60% glycolic acid units. There are significant shortcomings in this manuscript that need to be addressed. Therefore, paper must be improved significantly before reconsideration for publication.

·       Overall inconsistencies must be addressed spacing “ ” before units of measurement (either no spacing or spacing everywhere.

·       Typos and other mistakes should be checked out through the manuscript (for example page 1 line 35 “. [1-3]” should be “[1-3].” and page 3 “200°C- 230°C” should be “200°C – 230°C”), page 4 line 143 “um” should be “µm”, Figure 2 (x ass) spacing “Wavenumber(cm-1)” should be “Wavenumber (cm-1)”, page 6 line 202 “O=C-C-CH2” should be “O=C-C-CH2”, subscribt in page 8 for all the Tm, Tc etc. and so on.

·       In characterization section used equipment should be improved with information about the equipment production company and country.

·       Authors mention biodegradation even in abstract, but in the results and discussion there is only one reference to biodegradation and its water wetting angle.

·       Figure 2 range of 500 – 3200 cm-1 would be preferable as well as a rectangle instead of a square so that spectra would be clearer. A lot of differences can be missed if the spectra are so small.

·       Results and discussion section has only one reference (for equation), the discussion is missing explanation why one property is increasing or decreasing, why increasing glycolic acid (GA) more that 25 % results are contrariwise that up to 25 %, no comparison to other research works, while introduction section has discussed other cases of GA introduction in other biodegradable plastics, but no comparison of the results to this research. Discussion is “improved”, “increased”, “decreased” but by how much and why is not discussed.

·       Biodegradability results (at least initial results) should be added or at least some prediction based on water wetting results (by comparing it with other research containing water wetting and biodegradation).

Round 2

Reviewer 1 Report

The paper was corrected in accordance to my comments.It can be published in the present form.

Author Response

Dear reviewer:

Thank you for your  constructive comments.

Reviewer 2 Report

Authors have addressed most of the comments, but still some things coulb be improved:

Page 1 lines 24 and 28 O2 should be O2 in subscribt

Still some problems with spacings (Page 1 line 42 and further on) "performance[8-11]" same as double reference [8-11] is repeating twice.

Figure 1 (FTIR) Authors have indeed enlargened tha graph, nontheless, suggestion on cutting of the part where no information can be seen (3100-4000 cm-1) has not been implemented and the specra are still small. Y ass could be reduces (white regions) so that peak intensities are bigger and spectra could be strethced as wide as the text. Spectra of pure GA would be beneficial to se where changes would atributed to GA part and where to PBAT or some synergy. Peak shoulder for 1710 cm-1 is not discussed as well as intensity changes for 800 cm-1.

Subscribtions are not corrected everywhere: Page 5 line 159, 160, 161, 163, 164, 168, 188 etc. (for Hm, Tg, Xc...)

Reason for paper organization? Ussually for MDPI journals Results and discussion comes after Materials and methods section not before.
